# Corrosion Detection by Infrared Attenuated Total Reflection Spectroscopy via Diamond-Like Carbon-Coated Silicon Wafers and Iron-Sensitive Dyes

**DOI:** 10.3390/s19153373

**Published:** 2019-08-01

**Authors:** Dervis Türkmen, Carina Dettenrieder, Pontus Forsberg, Andreas Mattsson, Fredrik Nikolajeff, Lars Österlund, Mikael Karlsson, Boris Mizaikoff

**Affiliations:** 1Institute of Analytical and Bioanalytical Chemistry; Ulm University, 89081 Ulm, Germany; 2Department of Engineering Sciences, The Ångström Laboratory, Uppsala University, P.O. Box 534, SE-75121 Uppsala, Sweden

**Keywords:** mid-infrared chemical sensor, chemosensor, corrosion detection, diamond-like carbon, DLC, iron-sensitive dye, attenuated total reflection

## Abstract

The durability of metal-based constructions, especially those containing reinforced concrete, is mainly limited by corrosion processes. Diamond-like carbon (DLC)-coated silicon (Si) wafers provide a chemically inert and mechanically robust sensing interface for application in aggressive environments. In this study, iron-sensitive dyes, i.e., 2,3-dihydroxypyridine (DHP) and 1,2-dihydroxybenzol (DHB), were coated onto DLC-modified Si wafers for evaluating the potential of detecting corrosion processes via evanescent field absorption spectroscopy using Fourier-transform infrared spectroscopy. The obtained IR spectra reveal discernible changes of the dye layer after exposure to iron solutions, which indicates that indeed corrosion processes may be studied at molecular level detail.

## 1. Introduction

Iron ore mining worldwide amounts to approximately 1670 million tons and is mainly used for the production of iron and other important materials [1]. Iron is used to build cars, ships, bridges, and many other necessary objects, as well as buildings, pushing mankind further and further towards a modern future. The biggest advantage of using iron and its variations is usually the same: high load-bearing capacity combined with a high tensile strength despite high ductility, as is known, for example, from bridges, make this raw material versatile [2,3]. 

However, all iron end products share a common problem; corrosion. In the presence of water and oxygen, Fe(0) oxidizes to Fe(II). The second step of oxidation is the formation of Fe(III). The different oxidation states of the iron ions give rise to the later main components of rust. Therefore, Fe(II) and Fe(III) hydroxide convert to Fe(II) and Fe(III) oxide to form $x\mathrm{FeO}\cdot y{\mathrm{Fe}}_{2}O_{3}\cdot zH_{2}O$ [4] and, by forming an iron-damaging layer, lead to damage in the billions every year and can represent a safety risk [5,6,7].

Exactly this has to be recognized early in order to be able to intervene at the point where it is necessary to be able to prevent safety risks and consequential damages. For this reason, the development of novel sensors in this area is indispensable. However, the development of such sensors is strongly limited by environmental influences and corrosion-sensitive environments. Previous investigations of Graham and Cohen [8] show analysis methods of iron corrosion products using the Mössbauer spectroscopy. These methods show the possibility to use low energetic γ-rays combined using the Doppler effect mainly on solid iron products. The limitation of this spectroscopy is the laboratory use only due to complex equipment and its high costs. Moreover, when measuring liquid samples, inaccurate results are obtained compared to solid samples. 

Alternatively to Mössbauer spectroscopy, electrochemical impedance spectroscopy (EIS) can be applied using a certain voltage and current to, e.g., a metal sheet covered in an aqueous solution. To perform a measurement a three-electrode (working electrode, counter electrode, and a reference electrode) cell is necessary. Due to its various evaluation methods the EIS is very time-consuming and needs also good knowledge in electrical engineering. Furthermore, due to its sensitivity to vibration and more, this method is largely limited to measurements in the laboratory [9,10,11]. Attenuated total reflection Fourier-transform infrared spectroscopy (ATR-FTIR) was used within this study to analyze dye-coated diamond-like carbon (DLC) samples utilizing a diamond attenuated total reflection (ATR) module. Attenuated total reflection infrared (ATR-IR) spectroscopy is based on the formation of an evanescent field penetrating into the adjacent medium. Molecules of compounds within the penetration depth can be detected. The principle of ATR-IR spectroscopy can be found in detail elsewhere [12].

Si wafer chips coated with DLC are mechanically and chemically inert, have a high refractive index, and a broad optical transparency in the mid-infrared (MIR) region offering a wide application in sensing systems [13,14,15,16,17], e.g., in-field deployment as an optical transducer within an aggressive environment, i.e., being exposed to high temperature and humidity changes.

Fe detection is usually performed via color changes based on complexation reactions. Therefore, special compounds, such as 2,3-dihydroxypyridine (DHP), 1,2-dihydroxybenzol (DHB), potassium ferrocyanide and ferricyanide, thioglycolic acid, and thiocyanate can be used. With ultraviolet-visible (UV/Vis) spectroscopy color changes in the visible spectral region can be observed. However, for real-world applications the sensor may be built inside an entire system to detect corrosion processes of inner parts, i.e., steel ropes in wind power plants. Therefore, DHP and DHB were chosen within this study based on IR signature changes due to structural changes by addition of Fe ions.

In this study, ATR-IR spectra were recorded in order to detect Fe ions, and therefore, corrosion processes. Si wafers were coated with DLC and, subsequently, with a dye sensitive to Fe ions. DHP and DHB were investigated for this purpose due to direct chemical reaction even with low amounts of Fe. Changes of the IR spectra verified the presence of Fe ions. A linear calibration of DHP was established revealing a detection limit of 0.013 ± 0.005 mg/µL. Therefore, a sensor system based on Fe-sensitive dyes coated onto DLC chips can be implemented for corrosion detection within sites where corrosion leads to safety risks.

## 2. Materials and Methods

### 2.1. Materials

DHP (95%), DHB (99%), 2-propanol (99.9%), and ethanol (99.8%) were obtained from Sigma-Aldrich Chemie GmbH (Munich, Germany). Fe(III) chloride (98%) was supplied from Merck KGaA (Darmstadt, Germany)

### 2.2. Sample Preparation

Si wafers (10.16 cm in diameter) were cleaned with standard RCA 1 and 2 procedure, and subsequently dipped in HF:water (1:50) solution to remove surface oxides. Subsequent to the oxide removal, the wafer was placed in a deposition chamber for pump-down to a base pressure of 10^−5^ Pa. Using pulsed filtered cathodic arc deposition with a high-purity graphite cathode, a carbon-based coating was deposited to a total thickness of 30 nm at an electrical bias of −40 V. The first half of the coating thickness was deposited at base pressure to yield a regular tetragonal amorphous carbon (ta-C, i.e., DLC) coating, and then, for the top half of the coating, SF_6_ was introduced into the deposition chamber at a pressure of 1 Pa to yield a highly hydrophobic fluorinated amorphous carbon (F:a-C) top layer. The chamber was then vented to atmosphere with nitrogen. The surface-terminated DLC chips were diced into 1 × 1 cm² segments. Afterwards, the samples were cleaned with high purity 2-propanol. The respective dye was dissolved in pure ethanol. The DLC surface was coated with 50 µL of solute dye by using an Eppendorf pipette. To ensure complete removement of the solvent, the chips were dried for 24 h at room temperature, followed by a subsequent drying process for 12 h at 60 °C. For detecting Fe ions, Fe(III) chloride was dissolved in ethanol in the respective concentration. 10 µL of the Fe solution was dip-coated onto the dye-coated chip using an Eppendorf pipette. The prepared chip was dried for subsequent measurements.

### 2.3. Measurement Parameters

IR absorption spectra were recorded on a vacuum-pumped Bruker IFS 66v/S FT-IR spectrometer (Bruker, Ettlingen, Germany) equipped with the DuraSamplIR II ATR module with a diamond crystal as internal reflection element (IRE) (SensIR technologies, Danbury, CT, USA) and a broadband liquid nitrogen-cooled MCT (mercury cadmium telluride) in the 4000 to 600 cm^−1^ spectral region. Three-hundred scans were averaged for each spectrum with a spectral resolution of 2 cm^−1^. The dye-coated DLC chips were pressed against the diamond IRE crystal according to a procedure described elsewhere [18]. All measurements were performed at room temperature. The obtained IR spectra were baseline corrected by using the software Essential FTIR.

## 3. Results

First, the pure DLC chip was measured with an air background spectrum as shown in Figure 1. The absorption features from 4000 to 3500 cm^−1^ and 1800 to 1300 cm^−1^ (O–H stretching and bending vibration, respectively) are negative rotational–vibrational bands due to water vapor from the air background measurement which is not present during sample, i.e., pure DLC, measurement. Hydroxylic groups from water adsorb over time to DLC causing a strong IR band from 3550 to 3050 cm^−1^ due to the O–H stretching vibration. Other absorption bands from DLC or impurities do not occur. Therefore, interferences from DLC are reduced, since characteristic absorption peaks occur in the fingerprint region, i.e., below 1500 cm^−1^. The signal-to-noise ratio was calculated to be 1.7753.

Based on their application as an iron detection agent, DHP and DHB were chosen as dye to be placed on the DLC chip. Figure 2 illustrates the chemical structures of DHP and DHB. In the visible spectral range, a color change from colorless to gray, and colorless to blue for DHP and DHB, respectively, can only be observed by adding higher amounts of Fe ions. Furthermore, the concentration of the dye has to be high. Therefore, application of UV/Vis spectrometry is difficult. For a fast detection of corrosion processes with even small amounts of Fe, installed FT-IR sensors may overcome this problem. Therefore, differences in the IR absorption spectrum due to chemical structure changes should appear, if corrosion occurs.

The red MIR spectrum (Figure 3) illustrates DHP applied to the DLC chip compared to DHP measured in solid state (black). The IR signature of both spectra are similar and consistent to the literature [19]. IR peaks below 1800 cm^−1^ are higher in solid state due to a higher sample concentration. When applied to DLC, the concentration of dye is decreased. The broad band at 3260 cm^−1^ corresponding to hydroxylic groups of DHP applied to DLC indicating the formation of associates, is shifted to lower wavenumbers together with a peak widening (~3100 cm^−1^) if DHP is in solid state due to strong hydrogen-bridge bonds. Hydrogen-bridge bonds further lead to a decrease in intensity. By application to DLC, hydrogen-bridge bonds do not occur and the intensity of the O-H stretching vibration is increased. Hence, the absorption intensity of O–H vibration from DHP on DLC appears to be the same compared to solid state [20]. Since there were no structural changes, the binding to DLC is based on physical bonding, i.e., adhesion forces, instead of chemical bonding.

During corrosion processes, Fe ions are produced. For this reason, 10 µL of a Fe solution with a concentration of 9.44 mg/mL was added to the DHP-DLC chip (Figure 4, red). For comparison, the IR spectrum before iron addition is shown (black). The pure DLC chip (Figure 1) was taken as background spectrum. Figure 4b illustrates an enlarged region of the characteristic absorption bands from 1700 cm^−1^ to 1000 cm^−1^, which are labeled for clarity. The bands observed at 3260 cm^−1^ (O–H stretching vibration,) 1672–1657 cm^−1^, 1608 cm^−1^, 1576 cm^−1^, and 750 cm^−1^ assigned to C–C stretching and bending vibration of aromatics, the IR feature at 1367 cm^−1^, corresponding to O–H deformation vibration [19] disappear by addition of iron ions suggesting the deprotonation of the hydroxylic groups due to coordination to iron and loss of aromaticity due to protonation of nitrogen. This band was used for further quantitative analysis and is labeled for clarity in the Figure 4. The new peak at 3441 cm^−1^ attributed to N-H stretching vibration supports this consideration. The peaks at 1608 cm^−1^ and 1576 cm^−1^ from aromatic stretching vibrations are shifted to lower wavenumbers due to weakening of bonds based on the additional bond of oxygen to iron.

In addition to DHP, DHB, acting as iron reagent dye, was applied to the DLC chip. After the drying process, an IR spectrum of DHB was recorded with the pure DLC chip as background (Figure 5, black). The IR signature of DHB onto DLC is similar to the literature [21]. Broad double peaks at 3445 cm^−1^ and 3318 cm^−1^ corresponding to hydrogen-bonded O–H vibration of the two hydroxylic groups are clearly apparent. The peaks between 1619 cm^−1^ and 1468 cm^−1^ are due to aromatic ring C=C vibrations. The absorption bands at 1279 cm^−1^ and 1256 cm^−1^ are ascribed to C–O stretching vibration. The out-of-plane =C–H vibration is at 849 cm^−1^ and 740 cm^−1^. The red line of Figure 5 represents the absorption spectrum after adding Fe ions to the DHB-DLC chip. Hence, if corrosion takes place, the two peaks from hydrogen-bonded O-H almost disappeared suggesting deprotonation of DHB and coordination to iron. Makuraza et al. [22] computed the spectrum of 1,2-benzochinone and the most intense IR feature was assigned to the C=O stretching vibration at 1734 cm^−^^1^. However, this peak was not observed after addition of Fe ions corroborating the coordination to iron. Furthermore, three new peaks at 2851 cm^−^^1^, 2919 cm^−^^1^, and 2952 cm^−^^1^ appeared, which might be correlated to sp^2^-hybridized C-H stretching vibrations suggesting that protons from the hydroxyl groups were attached on the cyclic ring forming a cyclohexene ring. Additionally, IR bands from the aromatic system disappeared. The absorption feature from C–O stretching vibration (1279 cm^−^^1^ and 1256 cm^−^^1^) shifted to lower wavenumbers due to weakening of the C-O bond based on the coordination to iron, i.e., formation of a new bond.

Both reagents, DHP and DHB, revealed IR spectroscopic changes due to structural changes after adding Fe ions. The absorption features of both dyes indicate the coordination to iron via oxygen bond. Therefore, their application for iron detection was successfully demonstrated. While DHP poses no threat to the ecosystem and can be used without hesitation, DHB is toxic and in part mutagenic. Hence, environmental deployment is difficult and discharges into the environment have to be prevented.

Therefore, quantitative analysis of iron with DHP dye was performed. For this, the IR absorption feature at 1367 cm^−1^ was used for evaluation. If Fe ions from corrosion process interact with the dye, the hydroxylic group is deprotonated due to coordination to Fe, therefore a decrease of the band was observed. Figure 6a shows the IR spectra of the respective concentration used for establishing the calibration function (Figure 6b), which was obtained from the peak area vs. the concentration of iron solution added to DHP-DLC chip. The concentration ranges from 0.063 mg/µL to 0.315 mg/µL. The error bars were from five repititive measurements. The obtained goodness of fit (r^2^-value) was 0.99668. The limit of detection (LOD) and limit of quantification was calculated according to the IUPAC 3σ- and 10σ-criteria (standard deviation of blank measurement), respectively. Hence, the LOD and LOQ was calculated to be 0.013 ± 0.005 mg/µL and 0.043 ± 0.004 mg/µL.

## 4. Conclusions

Evanescent field absorption spectroscopy for detection of Fe ions occurring from corrosion processes via dye-coated Si wafers with DLC coating has been successfully demonstrated. The dyes undergo a chemical reaction, i.e., coordination to Fe ions, and therefore a change in the IR absorbance spectra was observed, i.e., the O–H stretching vibration of DHP and DHB disappeared. In the case of DHB, coordination to iron leads to the loss in aromaticity, and formation of sp^2^ hybridized C–H vibrations was observed. Both, DHP and DHB revealed fast response characteristics once Fe ions were added. Exemplarily, a calibration function of DHP was established with a resulting detection limit of 0.013 ± 0.005 mg/µL. Due to direct chemical reaction with iron, the onset of corrosion processes can be detected and future integration as a mechanically robust corrosion detection system is enabled. Further studies are required to test the dye-coated Si wafer-based sensor behavior in harsh environments, e.g., changing weather conditions. Hence, long-term stability tests with changing temperature and humidity conditions have to be carried out. Based on the obtained results with DLC coated Si wafers, future application of DLC waveguides replacing the commercial IRE together with quantum cascade lasers may lead to higher sensitivities.

## Figures and Tables

**Figure 1 sensors-19-03373-f001:**
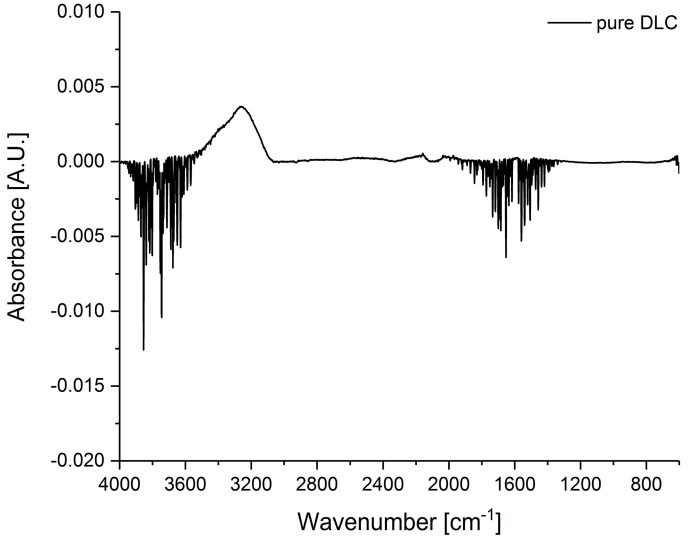
MIR spectrum of a cleaned pure diamond-like carbon (DLC) chip with air as background spectrum. Only absorption bands from water are visible.

**Figure 2 sensors-19-03373-f002:**
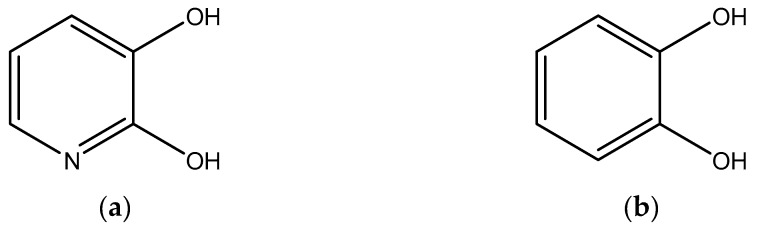
Chemical structure of iron sensitive dyes: (**a**) DHP and (**b**) DHB.

**Figure 3 sensors-19-03373-f003:**
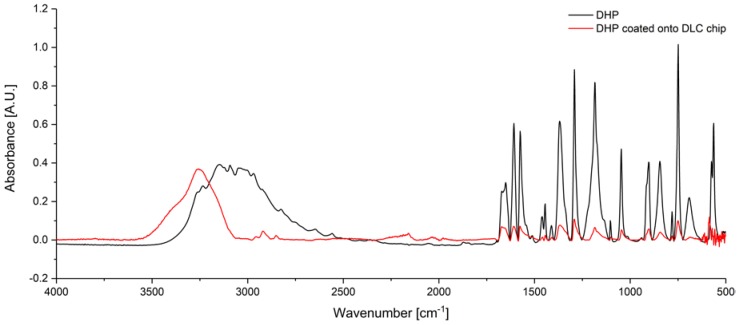
MIR spectrum of solid DHP (black) and 13 mg/mL DHP onto DLC layer (red).

**Figure 4 sensors-19-03373-f004:**
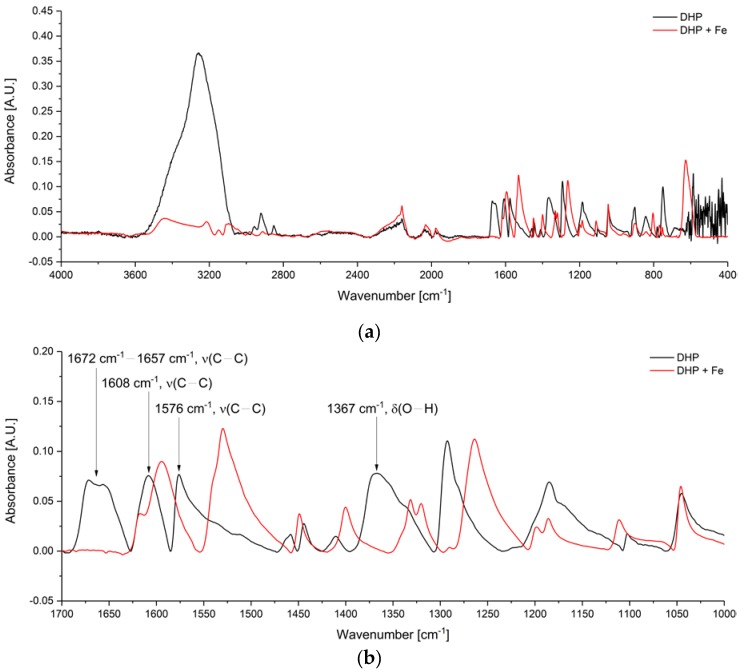
(**a**) MIR spectrum of DHP-coated DLC chip before (black) and after (red) adding 10 µL of 9.44 mg/mL solution Fe ions. The O–H stretching vibration from the hydroxylic groups and the C-C stretching vibrations disappeared due to coordination to Fe; (**b**) Enlarged region of characteristic absorption features from 1700 cm^−1^ to 1000 cm^−1^. The IR peak at 1367 cm^−1^ was evaluated for quantification.

**Figure 5 sensors-19-03373-f005:**
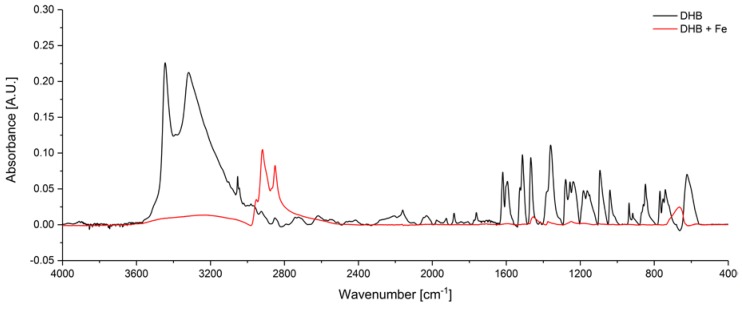
MIR spectrum of DHB before (black) and after (red) addition of Fe ions (10 µL with a concentration of 9.44 mg/mL). The absorption features of the hydroxylic groups from DHB and the aromatic ring vibrations decreased with Fe ions addition. New peaks from 2950 to 2850 cm^−1^ indicate the formation of CH_2_ within the cyclic ring.

**Figure 6 sensors-19-03373-f006:**
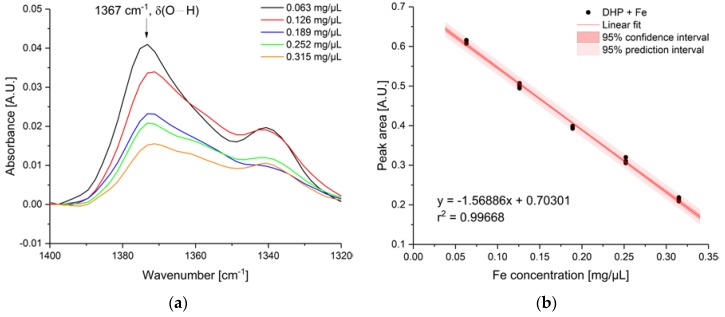
(**a**) IR spectra of the decreasing IR absorption feature of DHP from O–H deformation vibration at 1367 cm^−1^ used for quantitative evaluation; (**b**) Established linear calibration function. Five repetitive measurements of each Fe concentration were recorded in the range from 0.063 mg/µL to 0.315 mg/µL.

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
