# Peer review of "Corrosion Detection by Infrared Attenuated Total Reflection Spectroscopy via Diamond-Like Carbon-Coated Silicon Wafers and Iron-Sensitive Dyes"

_sensors, 2019, doi:10.3390/s19153373_

Round 1
Reviewer 1 Report
This paper investigates, by means of ATR spectroscopy, the Fe-sensitivity of two dyes, DHP and DHB, coated on DLC prepared on Si. The paper is essentially interesting and the results seem to be sound. At least to the reviewer, however, the description of the experiment was somewhat confusing. Please consider the following points.
(1) As I understood, a commercial IRE was used in ATR, and the surface of the DLC chip coated with dye was pressed on it. If this is the case, although the authors call it "DLC waveguide chip", it is not used as a waveguide, and the reviewer couldn't understand the role of the DLC layer. Is it planned to be used as a waveguide, replacing the IRE, in the future? If this is the scenario, the purpose of experiments in this paper should be explained more clearly.
(2) The procedure of "adding Fe ions" is not described. Please write in detail how it was done.
(3) The first sentence of 5. Conclusions is ungrammatical.
(4) Use a consistent name, either 2-propanol or isopropanol.
Author Response
(1) The terms of DLC waveguide were replaced by DLC coated Si wafer, the role of DLC was investigated for future application as waveguide.
(2) The procedure of adding Fe ions was added to the experimental section.
(3) First sentence of 5. Conclusion was grammatically revised.
(4) Isopropanol was changed to 2-propanol to be consistent.
Reviewer 2 Report
The paper provides a method to detect the corrosion processes via evanescent field absorption spectroscopy using Fourier transform infrared spectroscopy. Measured absorption signatures show that the detection is effective and the method is simple to understand. There are a few points that need to be clarified.
Diagrams of the measuring system and the detailed picture of the sensing head showing the relative position of each layer are necessary so as to better understand the mechanism of the sensing.
line108
"in figure 1 The water absorption features from 3550-3050 cm-1 and 1800-1300 cm-1. "
Why are the absorptions of positive and negative values, respectively? What is the absorption features from 4000-3500 cm-1?
line114.
"However, for detecting corrosion processes in systems, where visual detection is impossible,"
Comments are needed for the point. Is it possible to use a portable spectrometer in visible region to measure the color changes? It must be less expensive and convenient than an FTIR.
line125
"Fig 3" Solid state DHP has strong absorption in 1700-12500px-1 and around 75000px-1.
When DHP (maybe a DHP film) applied to the DLC chip, the absorption intensity around 75000px-1 is the same while the absorptions in 1700-12500px-1 become much weaker. Comments are needed on the phenomena.
line127-136
It is hard to find corresponding peaks in figure 4 in the explanation. An enlarged figure in 1600-10000px-1 is needed to show details and important absorption peaks should be labeled.
Author Response
- A new reference was cited in the experimental section showing the measurement principle.
- Water absorption from 4000-3500 cm-1 and 1800-1300 cm-1 are due to water vapor from ambient air, they are negative due to higher amounts of water vapor during background measurement compared to sample measurement. The absorption feature from 3550-3050 cm-1 is based on O-H stretching vibration from hydroxylic groups adsorbing to DLC surface over time.
- For detection of Fe ions via UV/Vis spectroscopy, high amounts of Fe and dye are needed to reveal a color change which is detectable with this method. For a fast detection of corrosion, only few amounts of Fe should be detected, which might not lead to a detectable color change, but to a change in the chemical structure, therefore, enabling detection via IR spectroscopy.
- Figure 3: The absorption intensity of solid state below 1800 cm-1 is increased compared to DHP on DLC due to higher concentration of DHP when measured in solid state. When applied to DLC, hydrogen-bridge bonds do not occur. Therefore, the absorption intensity is higher compared to solid state. Strong hydrogen-bridge bonds lead to a decrease in intensity with a shift to lower wavenumbers and a peak broadening. As a result, the intensity of O-H stretching vibration features the same intensity.
- In Figure 4, an enlarged figure (1700-1000 cm-1) was added, important IR peaks were labeled.
Round 2
Reviewer 2 Report
The authors have improved the paper. It can be accepted in present form.